Determination of natural populations to be included in breeding program in St. John’s wort species (Hypericum perforatum L.)

Uysal Bayar Fatma uysal.fatma@tarimorman.gov.tr
Medicinal Aromatic Plants Section, Bati Akdeniz Agricultural Research Institute , Antalya , Turkey
Fenu Giuseppe
Electronic publication date: 2024 Oct 16
Publication date: 2024
Volume: 12
Electronic Location ID: e18336
Received 2024 May 31; Accepted 2024 Sep 25
Copyright: © 2024 Uysal Bayar
Copyright year: 2024
Copyright holder: Uysal Bayar
License: This is an open access article distributed under the terms of the Creative Commons Attribution License, which permits unrestricted use, distribution, reproduction and adaptation in any medium and for any purpose provided that it is properly attributed. For attribution, the original author(s), title, publication source (PeerJ) and either DOI or URL of the article must be cited.
License URL: https://creativecommons.org/licenses/by/4.0/

Keywords: St. John’s wort, Selection, Cultivar, Yield, Quality

Funding: Turkish Ministry of Agriculture and Forestry, General Directorate of Agricultural Research and Policies TAGEM/TBAD/B/20/A7/P6/5286 This work was supported by the Turkish Ministry of Agriculture and Forestry, General Directorate of Agricultural Research and Policies, Project No. TAGEM/TBAD/B/20/A7/P6/5286. The funders had no role in study design, data collection and analysis, decision to publish, or preparation of the manuscript.

==============================
St. John’s wort (Hypericum perforatum L.) is a medicinal and aromatic plant of rapidly increasing importance. The cultural production of this species, which is of economic importance due to its medicinal properties, is limited. One of the main ways to increase production is to develop cultivars. Thus, the homogeneous raw material required for a standard product will be provided. This study aimed to determine the characteristics of natural populations to obtain productive cultivars with high hypericin and hyperforin that can meet market demands. In addition to yield and quality values, other characteristic features of the plant such as flower diameter, petal length and petal width, which directly affect productivity, were revealed in the study. The study was conducted under field conditions for two years with three replications. Fresh flower weight among the populations varied between 30.15 and 240.28 g/plant, while the hypericin ratio varied between 51.32 and 105.31 (mg/100 g). The study determined a wide variation among the populations, and the populations with superior characteristics were included in the breeding program.

Introduction

Hypericum L. is the type and the largest genus of the family Hypericaceae, including 420–470 species, mostly herbaceous plants, shrubs, or rarely small trees or annual species classified into 30–36 sections according to the most recent reviews (Crockett & Robson, 2011). Depression is a neuropathological disorder that affects 3.8% of people. St. John’s Wort (Hypericum perforatum L.) is a popular dietary supplement ingredient used in some countries to treat mild to moderate depression. Recent pharmacological studies report that St. John’s Wort extracts are effective in treating depression due to the presence of hypericin and hyperforin (Lou et al., 2022; Wang et al., 2023). With the increasing knowledge about the pharmacological activity of plant extracts, hypericin, hyperforin, and similar compounds, interest in St. John’s wort has increased. For this reason, St. John’s wort has become one of the most researched medicinal plants in the last two decades (Kaplan et al., 2023). These studies have generally focused on the chemical content of the plant collected from the natural flora and the pharmacological and clinical properties of the extracts.

With the increasing knowledge about the pharmacological activity of plant extracts, hypericin, hyperforin, and similar compounds, interest in St. John’s wort has increased. For this reason, St. John’s wort has become one of the most researched medicinal plants in the last two decades (Kaplan et al., 2023). These studies have generally focused on the chemical content of the plant collected from the natural flora and the pharmacological and clinical properties of the extracts.

Although this medicinal and economic significant plant is widespread in the natural flora, its production under cultivated conditions is limited. The available literature shows that there is very little research on evaluating the bio-agronomic and phytochemical response of H. perforatum to open field conditions. Most articles are based on plant specimens collected from nature or plants grown under restricted conditions, mostly in pots (Lazzara, Carrubba & Napoli, 2021).

In recent years, a shortage of standard products has emerged with the increase in demand for natural products. These standard products can only be provided with standard materials. The main way to obtain standard material is to develop the right cultivation. Some cultivars such as HyperMed, Elixir, Topaz, and others have been developed and traded. The most widely used of these is the Topaz cultivar, developed in 1982 (Pospielov, Pospielova & Semenko, 2023).

The quality of St. John’s wort may vary according to subspecies or cultivars, geographical characteristics of the region, and harvest time (Verotta, 2003; Lazzara, Carrubba & Napoli, 2021). Environment, genotype, or their interactions also affect the plant’s chemical composition (Bagdonaitė et al., 2010). Significant genotypic and phenotypic differences can occur between populations (Walker et al., 2001). Therefore, it is essential to identify genetic diversity and perform morphological and biochemical characterizations to develop region-specific cultivars (Kaplan et al., 2023).

St. John’s wort is a species whose flowers are used. Therefore, the most important observations affecting flower yield are observations related to flowers. Flower size and density are the most important features affecting flower yield. Therefore, flower diameter data were obtained in the study. Since the flower diameter feature will also be affected by petal measurements, petal width and petal length measurements were made. In addition, the number of flowering shoots, the distance between the buttonhole and the last flower affect flower density. In addition, when St. John’s wort is harvested, approximately 1/3 of the plant is harvested and these parts contain leaves. In general yield values, leaves are located together with flowers. Large leaves are an undesirable situation as they will reduce the flower ratio in general yield values. In other words, selecting large flower and small leaf populations will be advantageous in terms of productivity. It is important to characterize the available material in order to select the most suitable one for the purpose from the variation created in breeding studies. In this study, it was aimed to obtain varieties with high flower yield, high hypericin and hyperforin content, suitable for regions with subtropical climate conditions such as Turkey. For this purpose, 18 populations collected from the natural flora of Turkey were grown under equal field conditions and characterized morphologically, and the prominent populations were included in the breeding program.

Materials and Methods

Materials

Hypericum perfaratum seeds collected from its natural flora were used as material. Topaz commercial cultivar was used as a control. Herbarium specimens of the material used in the study are kept in the National Gene Bank of the Aegean Agricultural Research Institute and the herbarium of Nezehat Gökyiğit Botanical Garden. The location information of the seeds collected from nature and the barcode numbers of the herbarium specimens are given in Table 1.

Table 1 Location information of seeds collected from nature.

Population code	Population name	Coordinate and registration information	Altitude (m)	Barcode no	
1-AL	Sapadere/Alanya	36°29′14″N 32°18′3″E	378	NGBB 018001	
4-KC	Cesme/Kas	36°22′28″N 29°43′52″E	536	NGBB 018002	
5-KGU	Üçoluk/Gömbe/Kaş	36°31′40″N 29°39′34″E	1,312	NGBB 018003	
6-EA	Between Gömbe-Akçay/Kaş	36°34′35″N 29°43′55″E	1,124	NGBB 018004	
7-EC	Cobanisa-Elmalı	36°51′59″N 30°1′43″E	1,188	NGBB 018004	
8-KK	Town/Kas	36°18′35″N 29°43′46″E	250	NGBB 018005	
9-AY	Altinyaka/Kumluca	36°33′42″N 30°20′41″E	926	NGBB 018006	
10-MB	Beskonak/Manavgat	37°10′11″N 31°11′11″E	166	NGBB 018007	
11-MC	Caltepe/Manavgat	37°18′44″N 31°12′8″E	485	NGBB 018008	
12-SH	Haskiziloren/Serik	37°29′14″N 30°98′32″E	990	NGBB 018009	
13-AMI	Murtiçi/Akseki	36°89′63″N 31°76′45″E	537	NGBB 018010	
14-GC	Rotten/Gundoğmuş	36°47′53″N 31°50′49″E	719	NGBB 018011	
15-GU	Umutlu/Gundoğmuş	36°46′56″N 32°0′30″E	886	NGBB 018012	
16-GPD	Doganca/Gazipasa	36°15′81″N 32°32′20″E	700	NGBB 018013	
17-MU	Mugla	36°96′60″N 28°68′86″E	704	NGBB 018014	
18-BAL	Balıkesir	39°38′54″N 27°52′57″E	470	TR 55375	
19-BIL	Bilecik	40°09′31″N 29°58′33″E	1,788	TR 82501	
20-KUT	Kutahya	39°25′00″N 29°58′60″E	1,108	TR 71546	

Study area and climatic parameters

The study was conducted from 2021 to 2023 in the field of Bati Akdeniz Agricultural Research Institute (36.56 °N, 30.53 °E, and altitude 28) in Turkey. Monthly average air temperature and precipitation values for the experimental years are given in Table 2. In the years 2022 and 2023, the total rainfall was about half of the long-year average. However, since the experiment area was irrigated in a controlled manner, the study was not negatively affected. There were no adverse effects on the other weather conditions that would affect the study.

Table 2 Meteorological data of the experiment years.

Months	Total precipitation (mm)	Mean temperature (°C)	
LYA*	2021	2022	2023	LYA*	2021	2022	2023	
January	234.6	317.0	168.8	20.6	10.0	11.2	10.62	10.04	
February	152.1	26.0	16.1	6.2	10.7	12.3	10.72	9.13	
March	94.0	35.0	125.0	215.0	12.9	12.6	13.22	13.83	
April	49.4	4.0	0.0	26.0	16.4	16.8	18.92	15.48	
May	32.1	5.0	7.6	173.2	20.6	22.3	20.96	19.43	
June	11.0	18.0	21.4	6.4	25.3	25.0	25.28	23.91	
July	4.5	0.0	0.0	0.2	28.5	29.7	28.76	30.03	
August	4.5	1.0	2.2	0.0	28.4	28.3	27.17	27.50	
September	16.6	24.0	0.0	7.2	25.2	24.7	24.22	24.86	
October	67.9	14.0	25.6	4.4	20.6	20.6	20.31	20.94	
November	132.1	382.0	89.8	0.03	15.5	17.6	15.19	20.71	
December	261.2	236.0	8.20	123.0	11.6	13.3	12.48	12.89	
Mean	–	–	–	–	18.8	19.5	18.99	19.06	
Total	1,060.0	1,062.0	464.7	582.23	–	–	–	–	
Note:

* LYA: long year average (1930–2022).

The soil in which the research was carried out is clay-loam, not salty, very high calcareous, and strongly alkaline. Furthermore, the soil has low organic matter content, high phosphorus, calcium, magnesium, moderate potassium, sufficient manganese, iron and copper, and insufficient zinc.

Planting and growing conditions

Seeds collected from different populations were sown in germination boxes under greenhouse conditions in October 2021. The obtained seedlings were planted in field conditions in the spring of 2022 in three replications with four rows in each replicate at a distance of 20 × 25 cm. The length of the rows was 2 meters and there were a total of 32 plants in each plot. Irrigation was done with a drip irrigation system when the soil moisture was below the field capacity. Perennial mulch material was used for weed control. No pesticides or fertilizers were used in cultivation. All populations were compared under equal conditions. Visuals of the field studies are given in Fig. 1. In addition to parameters such as plant height, stem number, stem diameter, leaf length, leaf width, distance between the highest and lowest flowers, number of flowering shoots, flower diameter, petal length, and petal width were determined before harvesting in accordance with UPOW criteria. Harvesting was done in early June when more than 50% of the buds had flowered. Harvesting was performed by cutting with scissors just below the first flowering (about 1/3 of the upper part of the plant). Fresh and dry flower weight was determined after harvesting.

Figure 1 (A) Seedling production in the greenhouse, (B) general view of the experimental area, (C) difference between populations.

Analysis methods

The drying process was performed in an oven at 40 °C for 2 days. The amounts of pseudohypericin, hypericin, and hyperforin were determined in the dried samples. Analyses were performed according to Isacchi et al. (2007). According to this method, a 1 g sample was subjected to extraction with methanol. The extract was then clarified in a cooled ultracentrifuge at 5,000 rpm for 5 min, passed through a 0.45 µm filter, and injected into LC/MS-MS. The active substance composition of the samples was analyzed on a C18 column (Zorbax SB-C18, HT 2.1 × 50 mm, 1.8 µm), and the column temperature was 35 °C. Water: methanol (A, 95:5, 5 mM ammonium formate, 0.01% formic acid) and methanol (B, 5 mM ammonium formate, 0.01% formic acid) were used as mobile phase and the flow rate was 0.3 mL/min. Active substance analysis conditions are given in Table 3.

Table 3 Pseudohypericin, hypericin, and hyperforin analysis conditions.

Duration (minutes)	A (%)	B (%)	
0.00	95	5	
0.50	95	5	
2.00	20	80	
5.00	10	90	
11.00	5	95	
12.00	5	95	
12.10	95	5	
13.00	95	5	

Statistical analysis was performed using the JMP software (SAS Institute, Cary, NC, USA). Means were compared by analysis of variance (ANOVA) and the LSD test at the p ≤ 0.05 significance level.

Results

The effects of different populations on plant height, stem diameter, leaf length, and leaf width were significant in both years. The effect of populations on stem number was significant in the first year but not significant in the second year (Table 4). The maximum, minimum and average plant height, stem diameter, leaf length and leaf width values are given in Table 5.

Table 4 Analysis of variance table for characteristics.

Parameters	Source of variation	Degrees of freedom	Mean squares	F value	P value	
Years		2022	2023	Mean	2022	2023	Mean	2022	2023	Mean	2022	2023	Mean	
Plant height (cm)	Block	2	2	2	297.41	297.41	90.89	5.60	5.60	3.32	0.0076	0.0076	0.0473	
Population	18	18	18	1,088.15	1,088.15	1,390.20	20.49	20.49	50.84	<0.0001	<0.0001	<0.0001	
Error	36	36	36	53.11	53.11	27.34							
Number of stems (number of plants )-1	Block	2	2	2	7.19	283.70	77.02	0.23	2.43	2.12	0.79	0.1028	0.1350	
Population	18	18	18	58.91	201.23	65.30	1.92	1.72	1.80	0.046	0.0816	0.0664	
Error	36	36	36	30.64	116.97	36.36							
Stem diameter (mm)	Block	2	2	2	2.50	0.23	0.44	5.25	1.02	2.07	0.0100	0.3701	0.1415	
Population	18	18	18	1.58	3.42	1.76	3.32	15.21	8.26	0.0011	<0.0001	<0.0001	
Error	36	36	36	0.48	0.22	0.21							
Leaf length (mm)	Block	2	2	2	9.50	0.86	1.47	1.86	0.13	0.45	0.1700	0.8808	0.6441	
Population	18	18	18	13.98	26.49	12.35	2.74	3.95	3.75	0.0049	0.0002	0.0004	
Error	36	36	36	5.10	6.71	3.30							
Leaf width (mm)	Block	2	2	2	1.20	0.60	0.77	0.32	1.66	0.72	0.7253	0.2040	0.49	
Population	18	18	18	4.36	1.63	1.37	1.17	4.51	1.29	0.3302	<0.0001	0.2509	
Error	36	36	36	3.71	0.36	1.06							
Distance between the highest and lowest flowers (cm)	Block	2	2	2	14.78	9.08	3.48	0.92	1.86	0.60	0.4079	0.1699	0.5559	
Population	18	18	18	83.37	33.01	45.97	5.19	6.77	7.89	<0.00001	<0.0001	<0.0001	
Error	36	36	36	16.07	4.88	5.83							
Number of flowering shoots (Number)	Block	2	2	2	4.35	1.67	2.74	0.61	0.55	1.38	0.55	0.5844	0.2655	
Population	18	18	18	16.60	9.81	9.40	2.35	3.20	4.72	0.01	0.0015	<0.0001	
Error	36	36	36	7.07	3.07	1.99							
Flower diameter (mm)	Block	2	2	2	2.97	11.34	5.44	0.94	2.84	2.58	0.4001	0.0713	0.0900	
Population	18	18	18	5.95	20.90	7.44	1.88	5.24	3.52	0.0522	<0.0001	0.0006	
Error	36	36	36	3.16	3.99	2.11							
Petal length (mm)	Block	2	2	2	0.91	0.85	0.81	1.07	0.84	1.85	0.3544	0.4404	0.1716	
Population	18	18	18	2.64	2.79	1.38	3.11	2.76	3.18	0.0019	0.0046	0.0015	
Error	36	36	36	0.85	1.01	0.44							
Petal width (mm)	Block	2	2	2	0.65	0.31	0.20	1.67	0.47	0.63	0.2033	0.6300	0.5399	
Population	18	18	18	0.89	3.33	1.40	2.29	4.98	4.47	0.0169	<0.0001	<0.0001	
Error	36	36	36	0.39	0.67	0.31							
Fresh flower weight g/plant	Block	2	2	2	748.78	918.37	618.54	8.02	0.56	1.65	0.0013	0.5740	0.2058	
Population	18	18	18	3,045.30	1,9817.48	9,770.36	32.63	12.17	26.10	<0.00001	<0.0001	<0.0001	
Error	36	36	36	93.33	1,628.79	374.36							
Dry flower weight g/plant	Block	2	2	2	65.55	152.71	71.52	4.49	0.48	0.94	0.0181	0.6255	0.4011	
Population	18	18	18	437.59	4,123.05	1,651.04	29.99	12.84	21.63	<0.00001	<0.0001	<0.0001	
Error	36	36	36	14.59	321.22	76.32							
Pseudohypericin (mg/100 g)	Block	2	2	2	300.23	143.09	85.55	6.62	0.72	1.68	0.0036	0.4937	0.1998	
Population	18	18	18	965.34	947.55	825.79	21.27	4.77	16.26	<0.00001	<0.0001	<0.0001	
Error	36	36	36	45.38	198.72	50.78							
Hypericin (mg/100 g)	Block	2	2	2	862.60	215.92	480.80	6.50	1.63	9.57	0.0039	0.2092	0.0005	
Population	18	18	18	288.53	1,265.20	564.73	2.17	9.58	11.21	0.0233	<0.0001	<0.0001	
Error	36	36	36	132.70	132.09	50.22							
Hyperforin (g 100 g −1)	Block	2	2	2	0.04	0.26	0.14	0.20	1.02	1.27	0.8221	0.3715	0.2943	
Population	18	18	18	1.00	1.22	1.02	4.67	4.75	9.24	<0.00001	<0.0001	<0.0001	
Error	36	36	36	0.21	0.26	0.11							

Table 5 Mean, minimum, maximum values and standard deviations of average plant height, stem number, stem diameter, leaf length and leaf width for individual levels.

Parameter/Traits	Plant height	Stem number	Stem diameter	Leaf length	Leaf width	
Mean	75.52	19.87	3.85	15.17	4.25	
Minimum	40.00	6.34	2.10	10.70	2.8	
Maximum	109.58	42.25	5.80	20.00	10.9	
Standard deviation	21.63	6.87	0.85	2.47	1.07	
Coefficient of variation (%)	28.64	34.56	22.09	16.31	25.12	
N	57	57	57	57	57	

Plant height varied between 50.50 and 105.67 cm in the first year and between 38.00 and 115.00 cm in the second year. In the first year of the study, the highest number of stems was obtained from the 19-BIL population, while in the second year, it was obtained from 12-SH. The populations responded differently in stem formation after mowing in the first year. While the 20-KUT population formed about four times more shoots after the first year of harvesting, the number of shoots was almost halved in the 8-KK population.

The stem obtained in all populations was thicker than the Topaz cultivar used as a control. Throughout the study, leaf length varied between 9.46 and 22.30 mm, while leaf width varied between 1.95 and 7.82 mm (Table 6).

Table 6 Plant height, stem number, stem diameter, leaf length, and leaf width data.

Population/
Parameter	Plant height (cm)	Stem number (Number/plant)	Stem diameter (mm)	Leaf length (mm)	Leaf width (mm)	
Year	2022	2023	Mean	2022	2023	Mean	2022	2023	Mean	2022	2023	Mean	2022	2023	Mean	
1-AL	103.33	110.83	107.08	15.33	22.78	19.06	4.94	5.58	5.26	16.04	22.30	19.17	4.41	5.20	4.81	
4-KC	57.67	63.17	60.42	10.50	32.83	21.67	3.59	3.82	3.705	12.65	20.31	16.48	3.55	5.21	4.38	
5-KGU	55.33	39.50	47.42	9.50	30.50	20.00	4.48	2.70	3.59	14.79	12.94	13.87	4.39	3.45	3.92	
6-EA	59.33	64.22	61.78	17.50	25.89	21.70	3.89	3.19	3.54	12.50	15.11	13.81	4.09	4.58	4.34	
7-EC	66.67	64.34	65.51	19.50	28.56	24.03	3.36	3.74	3.55	12.53	12.88	12.71	3.48	4.66	4.07	
8-KK	71.00	58.33	64.67	16.17	9.83	13.00	4.08	2.90	3.49	14.53	14.26	14.40	5.03	4.36	4.70	
9-AY	50.50	38.00	44.25	19.90	20.00	19.95	4.09	2.63	3.36	11.83	12.99	12.41	4.17	3.28	3.73	
10-MB	75.67	66.89	71.28	17.50	27.83	22.67	3.80	4.42	4.11	18.80	17.90	18.35	4.52	3.62	4.07	
11-MC	61.83	63.22	62.53	16.33	41.67	29.00	3.31	3.65	3.48	14.39	14.37	14.38	3.83	4.20	4.02	
12-SH	99.18	110.06	104.62	18.67	31.11	24.89	4.06	6.64	5.35	15.42	20.24	17.83	4.80	5.54	5.17	
13-AMI	84.83	80.00	82.42	14.50	27.78	21.14	3.67	4.05	3.86	14.91	13.34	14.13	3.34	3.78	3.56	
14-GC	75.17	76.39	75.78	18.17	25.17	21.67	3.97	4.57	4.27	15.29	18.33	16.81	3.71	4.20	3.96	
15-GU	68.33	59.72	64.03	18.17	17.00	17.59	3.54	2.90	3.22	16.76	14.42	15.59	3.16	4.42	3.79	
16-GPD	96.00	95.17	95.59	18.00	13.33	15.67	5.34	3.73	4.535	16.30	17.37	16.84	4.30	5.00	4.65	
17-MU	105.67	99.33	102.50	12.00	27.67	19.84	5.11	5.08	5.095	16.07	13.60	14.84	5.03	3.43	4.23	
18-BAL	102.33	115.00	108.67	20.67	22.33	21.50	3.74	5.03	4.385	12.87	17.30	15.09	7.82	4.64	6.23	
19-BIL	61.33	49.00	55.17	21.67	21.67	21.67	3.35	2.27	2.81	12.52	12.54	12.53	3.38	3.97	3.68	
20-KUT	95.67	111.00	103.34	6.33	23.33	14.83	3.90	5.39	4.645	13.54	18.83	16.19	2.62	5.91	4.27	
21-TOPAZ	52.67	62.64	57.66	8.00	7.67	7.84	2.17	2.07	2.12	9.46	16.06	12.76	1.95	4.41	3.18	

The effects of different populations on inflorescence length, number of flowering shoots, petal length, and petal width were significant in both years. While the effect of populations on flower diameter was not significant in the first year, it was significant in the second year (Table 4). The maximum, minimum and average inflorescence length, flower shoot number, petal length and petal width measurements are given in Table 7. Distance between the highest and lowest flowers varied between 8.67 and 27.00 cm in the first year and between 5.00 and 18.44 cm in the second year. In the second year, flowering length was shorter except for 8-KK and 10-MB in all populations. The number of flowering shoots is one of the traits affecting yield. While the highest number of flowering shoots was obtained from 12-SH in the first year, the highest number was obtained from 1-AL in the second year. While the largest flower diameter was obtained from 19-BIL in the first year, it was obtained from 18-BAL in the second year. In the first year, the flower diameter of all populations was larger than TOPAZ, which was used as a control. In the second year, 14 populations exceeded the control in terms of flower diameter. As expected, the longest and shortest petals were determined in the same populations in direct proportion to the flower diameter. Petal width varied between 4.87 and 6.89 mm in the first year and 2.82 and 5.99 mm in the second year (Table 8).

Table 7 Mean, minimum, maximum values and standard deviations of average distance between the highest and lowest flowers, number of flowering shoots, flower diameter, petal length and petal width for individual levels.

Parameter/Traits	Distance between the highest and lowest flowers	Number of flowering
Shoots	Flower diameter	Petal length	Petal width	
Mean	9.63	9.73	4.76	12.64	5.32	
Minimum	7.50	8.00	4.40	11.20	4.00	
Maximum	11.80	12.00	5.20	14.20	6.70	
Standard deviation	2.15	2.05	0.20	0.88	0.81	
Coefficient of variation (%)	22.32	21.09	4.28	6.93	15.31	
N	57	57	57	57	57	

Table 8 Distance between the highest and lowest flowers, number of flowering shoots, flower diameter, petal length, petal width data.

Population/
Parameter	Distance between the highest and lowest flowers (cm)	Number of flowering
Shoots (Shoots/Stem)	Flower diameter (mm)	Petal length (mm)	Petal width (mm)	
Year	2022	2023	Mean	2022	2023	Mean	2022	2023	Mean	2022	2023	Mean	2022	2023	Mean	
1-AL	21.33	14.33	17.83	10.67	14.22	12.45	23.77	24.09	23.93	13.14	13.37	13.26	6.25	5.83	6.04	
4-KC	11.17	8.11	9.64	7.00	8.11	7.56	23.78	24.79	24.29	13.85	13.42	13.64	6.38	5.85	6.12	
5-KGU	20.33	7.50	13.92	12.00	9.50	10.75	21.63	20.80	21.22	12.59	12.47	12.53	5.46	2.82	4.14	
6-EA	12.83	9.87	11.35	10.00	10.11	10.06	23.28	22.14	22.71	13.23	12.52	12.88	5.54	5.99	5.77	
7-EC	13.00	8.33	10.67	9.17	9.78	9.48	24.72	22.48	23.60	12.80	12.73	12.77	6.85	5.88	6.37	
8-KK	8.67	9.78	9.23	10.67	8.83	9.75	22.97	18.23	20.60	12.46	11.70	12.08	5.55	4.01	4.78	
9-AY	10.17	6.33	8.25	7.83	6.00	6.92	24.47	18.43	21.45	13.29	11.05	12.17	5.50	2.97	4.24	
10-MB	11.33	12.11	11.72	11.17	12.28	11.73	26.67	23.31	24.99	13.71	13.22	13.47	6.05	5.13	5.59	
11-MC	11.33	10.78	11.06	11.17	10.22	10.70	25.15	20.01	22.58	13.85	12.05	12.95	6.17	4.89	5.53	
12-SH	23.18	18.44	20.81	15.50	12.83	14.17	23.91	22.20	23.06	11.57	11.83	11.70	6.89	5.33	6.11	
13-AMI	14.83	10.33	12.58	12.00	8.67	10.34	25.93	21.18	23.56	13.93	12.58	13.26	5.69	4.78	5.24	
14-GC	15.17	10.06	12.62	10.33	10.06	10.20	24.99	19.47	22.23	13.74	10.79	12.27	5.92	5.01	5.47	
15-GU	17.33	13.39	15.36	14.00	10.11	12.06	24.45	15.46	19.96	13.17	11.12	12.15	6.21	3.37	4.79	
16-GPD	18.00	10.50	14.25	11.00	11.00	11.00	23.30	21.12	22.21	12.84	12.06	12.45	5.85	5.39	5.62	
17-MU	22.33	14.67	18.50	13.00	11.00	12.00	22.83	23.55	23.19	11.85	12.79	12.32	5.27	5.81	5.54	
18-BAL	21.00	15.33	18.17	13.00	10.67	11.84	23.21	26.03	24.62	11.39	13.35	12.37	5.10	5.22	5.16	
19-BIL	11.00	5.00	8.00	15.33	11.33	13.33	26.77	24.36	25.57	14.87	13.17	14.02	6.33	4.34	5.34	
20-KUT	27.00	12.33	19.67	7.67	11.67	9.67	23.80	21.72	22.76	12.17	13.05	12.61	5.88	4.84	5.36	
21-TOPAZ	13.00	10.19	11.60	11.33	10.36	10.85	21.57	18.98	20.28	11.65	10.17	10.91	4.87	3.02	3.95	

The effect of different populations on fresh flower weight, dry flower weight, pseudohypericin, hypericin, and hyperforin content was significant in both years (Table 4). The maximum, minimum and average fresh flower weight, dry flower weight, pseudohypericin, hypericin and hyperforin values are given in Table 9. The main objective in cultivation is to obtain a high yield. While the highest yield (130 g plant−1) was obtained from 20-KUT in the first year of the study, the highest yield (351.67 g plant−1) was obtained from 18-BAL in the second year. Fresh flower weight increased in the second year in all populations except 19-BIL.

Table 9 Mean, minimum, maximum values and standard deviations of average fresh flower weight, dry flower weight, pseudohypericin, hypericin and hyperforin for individual levels.

Parameter/Traits	Fresh flower weight	Dry flower weight	Pseudohypericin	Hypericin	Hyperforin	
Mean	86.63	46.43	70.95	66.17	1.77	
Minimum	73.50	12.00	38.00	47.00	1.30	
Maximum	102.20	105.60	123.90	111.00	2.10	
Standard deviation	14.50	24.13	17.35	15.19	0.18	
Coefficient of variation (%)	16.74	51.98	24.45	22.96	10.26	
N	57	57	57	57	57	

Quality is as important as yield in cultivation. Pseudohypericin, hypericin, and hyperforin are the most important factors affecting the quality of St. John’s wort. The highest population of pseudohypericin was determined in 7-EC in both years. Hypericin was the highest in 17-MU in the first year and 14-GC in the second year. The lowest hypericin rate was detected in 8-EC in both years. Hyperforin varied between 2.12 g 100 g−1 and 3.99 g 100 g−1 in the first year and between 2.11 g 100 g−1 and 4.35 g 100 g−1 in the second year (Table 10). Additionally, the hyperforin chromatogram of the standard sample and the real sample is given in Fig. 2.

Table 10 Fresh flower weight, dry flower weight, pseudohypericin, hypericin, and hyperforin data.

Population/
Parameter	Fresh flower weight (g plant−1)	Dry flower weight (g plant−1)	Pseudohypericin (mg/100 g−1)	Hypericin
(mg/100 g−1)	Hyperforin g 100 g−1	
Year	2022	2023	Mean	2022	2023	Mean	2022	2023	Mean	2022	2023	Mean	2022	2023	Mean	
1-AL	80.08	211.11	145.60	31.90	91.19	61.55	58.86	80.60	69.73	66.39	144.23	105.31	3.99	3.57	3.78	
4-KC	41.84	131.46	86.65	15.18	51.78	33.48	73.90	97.42	85.66	53.02	67.09	60.06	3.15	2.56	2.86	
5-KGU	11.30	49.00	30.15	4.21	22.83	13.52	72.42	82.33	77.38	49.03	55.96	52.50	3.42	3.23	3.33	
6-EA	52.10	101.35	76.73	19.81	40.14	29.98	82.19	97.05	89.62	55.89	71.35	63.62	3.74	3.89	3.82	
7-EC	42.59	88.96	65.78	15.28	33.52	24.40	104.25	124.10	114.18	68.02	66.90	67.46	3.68	4.20	3.94	
8-KK	50.10	184.11	117.11	20.94	88.48	54.71	60.46	76.83	68.65	46.72	55.92	51.32	3.48	3.78	3.63	
9-AY	15.01	40.97	27.99	5.65	18.50	12.08	61.63	54.02	57.83	56.35	65.37	60.86	2.57	2.11	2.34	
10-MB	60.77	208.70	134.74	23.36	86.00	54.68	58.70	69.74	64.22	66.12	68.23	67.18	2.50	2.84	2.67	
11-MC	55.85	202.23	129.04	21.09	88.37	54.73	67.29	64.05	65.67	50.92	56.63	53.78	2.36	2.39	2.38	
12-SH	69.19	252.50	160.85	26.80	111.25	69.03	71.31	93.97	82.64	52.58	73.27	62.93	3.19	3.35	3.27	
13-AMI	72.66	199.21	135.94	27.97	98.22	63.10	44.40	75.72	60.06	46.98	59.24	53.11	3.03	3.24	3.14	
14-GC	88.05	236.07	162.06	34.43	108.30	71.37	72.94	107.30	90.12	70.74	95.51	83.13	2.60	2.90	2.75	
15-GU	42.19	82.73	62.46	16.56	36.20	26.38	57.45	89.10	73.28	61.14	65.38	63.26	2.12	2.52	2.32	
16-GPD	81.93	152.95	117.44	31.38	69.45	50.42	44.99	78.77	61.88	53.42	73.43	63.43	3.89	4.35	4.12	
17-MU	91.00	160.90	125.95	35.81	66.53	51.17	54.75	83.99	69.37	81.90	90.45	86.18	3.54	3.60	3.57	
18-BAL	128.89	351.67	240.28	48.26	157.70	102.98	52.44	82.41	67.43	73.45	85.70	79.58	3.37	3.50	3.44	
19-BIL	41.84	31.47	36.66	5.65	10.61	8.13	35.42	48.98	42.20	54.35	56.68	55.52	2.18	2.31	2.25	
20-KUT	130.00	161.59	145.80	47.53	70.40	58.97	24.18	69.43	46.81	62.73	73.43	68.08	3.33	3.26	3.30	
21-TOPAZ	49.33	131.64	90.49	21.48	63.12	42.30	46.09	76.60	61.35	52.04	67.87	59.96	3.17	2.90	3.04	

Figure 2 The hyperforin chromatogram of the standard sample and the real sample.

Discussion

The plant height, stem number, and stem diameter values obtained in this study are similar to studies of Radušienė, Bagdonaite & Kazlauskas (2004), Riazi et al. (2011), and Lazzara, Carrubba & Napoli (2021). However, the upper value of the stem number was determined to be lower than the upper value of the study conducted by Riazi et al. (2011) in Iran. This situation may be explained by ecological or genotypic differences, as well as different cultural treatments or counting methods. The leaf width and length findings were similar to Riazi et al. (2011) but narrower and shorter than the values obtained by Radušienė, Bagdonaite & Kazlauskas (2004).

The data about flowers are in agreement with Radušienė, Bagdonaite & Kazlauskas (2004), Bagdonaitė et al. (2007), and Riazi et al. (2011). However, the distance between the highest and lowest flowers is considerably shorter than that obtained by Islam et al. (2021). Petal width was similar to Radušienė, Bagdonaite & Kazlauskas (2004), while different results were obtained for petal length. The mean petal length of the five genotypes used as material in the study was longer than that obtained by Radušienė, Bagdonaite & Kazlauskas (2004). Petal length affects flower diameter, which is one of the important parameters affecting flower yield. Since the most important feature affecting flower yield is flower diameter, flower diameter data were obtained. Since the flower diameter feature will also be affected by petal measurements, petal width and petal length measurements were made. In addition, when St. John’s wort is harvested, the flowering parts (approximately 1/3 of the plant) are harvested and this part also includes leaves. In general yield values, leaves are included with flowers. Large leaves are undesirable as they will reduce the flower ratio in general yield values. It is important to select large flower and small leaf populations.

Although the yield values of some of the genotypes in the study were similar to Pluhár, Bernáth & Németh (2002) and Lazzara, Carrubba & Napoli (2021), the yields of 1-AL, 8-KK, 10-MB, 11-MC, 12-SH, 13-AMI, and 14-GC were approximately two times higher. The yield obtained from 18-BAL was about three times higher.

Some of the results obtained in terms of pseudohypericin were similar to Pluhár, Bernáth & Németh (2002) but lower than Bagdonaitė et al. (2010) and Rychlewski et al. (2023). The hypericin content of the populations in the study was similar to Pluhár, Bernáth & Németh (2002), Bagdonaitė et al. (2007), Çirak et al. (2007), and Rychlewski et al. (2023). The 1-AL population with the highest average hyperforin ratio in the study was among the values obtained by Bagdonaitė et al. (2010). The hyperforin ratio obtained in the study is similar to that obtained by Butterweck et al. (2003) in aqueous ethyl alcohol extract. In the literature, the hyperforin ratio was generally determined in extracts, and no studies have determined it in plant material.

The different results obtained may be due to genetic and environmental factors, seasonal variations, growing conditions, post-harvest processing, analyzing methods, etc., Bagdonaitė et al. (2010) stated that the quality and quantity of components in H. perforatum depend on several factors. Cortés, López-Hernández & Blair (2022) and Cortés & Barnaby (2023) demonstrated that using innovative tools to accelerate prebreeding and discover adaptation sources by genotype from landraces, wild crop relatives and orphan crops is an important prerequisite to accelerate the genetic gain of abiotic stress tolerance. They also argued that there is a need to update current GEA (genome-environment associations) models to predict both regional and local or microhabitat-based adaptation with mechanistic ecophysiological climate indices and the latest GWAS-type genetic association models.

Conclusions

In conclusion, the study’s findings indicate that H. perforatum has great biochemical and morphological variability. The excess in plant biomass and flowering length in some populations is promising for breeding studies. Population number seven, which has approximately two times more pseudohypericin than the cultivar used as a control, and population number 14, which has 50% more hypericin content than the control, were included in the breeding study to be crossed with high-yielding genotypes.

Additionally, we believe that the results obtained in the study will aid future studies to combine evolutionary adaptive predictions with innovations in ecological genomics such as GEA, which uses such innovative approach models to capture latent genetic adaptations to abiotic stresses based on crop germ plasm resources to support responses to climate change.

Supplemental Information

Supplemental Information 1 Raw data.

Additional Information and Declarations

Competing Interests

Author Contributions

Data Availability

The author declares that they have no competing interests.

Fatma Uysal Bayar conceived and designed the experiments, performed the experiments, analyzed the data, prepared figures and/or tables, authored or reviewed drafts of the article, and approved the final draft.

The following information was supplied regarding data availability:

The raw data is available in the Supplemental File.

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
