# Peer review of "Determination of natural populations to be included in breeding program in St. John’s wort species (Hypericum perforatum L.)"

_PeerJ, doi:10.7717/peerj.18336_

## Round 0.1 · original submission · Major Revisions

Dear Author,
Two anonymous experts reviewed your article and highlighted several critical aspects of the MS that need revision.

Reviewer 1 ·

Basic reporting

1. Basic Report
Language:
The English used is generally sufficient to convey the informaon. Nonetheless, we recommend
improving the wording and adjusng the style to align with standard scienfic language. For example, in lines 29-32, the repeated use of the word "report" is not migated by using different tenses (past tense and present perfect progressive). Furthermore, expressions need correcon, such as in lines 85-86: "No ferlizaon or similar cultural pracces were applied, and all populaons were compared under equal condions." "Cultural" is not a term that can be used in this context. A be2er phrasing would be, "No ferlizaon or similar culvaon pracces were applied, and all populaons were compared under equal condions." Line 42-43 uses the adverb “economically” incorrectly. There are other expression and grammar errors, and we recommend revising the English language and scienfic style to improve the quality of the arcle. However, the informaon is clearly conveyed.
Intro, Background & Literature:
There are many very concise key statements about H. perforatum postulated in the introducon and
parally supported by literature. However, the flow of reading is disrupted by the lack of context.
There is no further explanaon or classificaon of the cited literature:
28 “Hypericum perforatum is a perennial herb in the Hypericaceae family. This species grows widely
29 in Europe, Asia, North Africa, and North America. It was reported that there are 89 species of
30 the Hypericum in Turkey, 43 of which are endemic (Cirak et al. 2016). It has been reported that
"Why is Turkey explicitly menoned here? Is Hypericum economically relevant for Turkey and if so,
how? Are there tradional applicaons in Turkey? Does Turkey export to other countries? Are the
menoned species relevant to the paper in any way? If so, briefly name and explain. Otherwise, the
other species can be omi2ed and the focus can be on Hypericum perforatum, or a general
introducon to the family and genus with their global representaves can be provided."
31 St. John's wort has an.depressant effects due to its hypericin content (Lou et al. 2022; Wang et
32 al. 2023). Moreover, the plant is seda.ve due to bioflavonoids and hyperforin (Mitchell 1999),
33 an.-inflammatory, an.ulcerogenic, analgesic, diure.c due to flavonoids and tannins
34 (Jahangirova et al. 2023), astrengenic, and an.phlogis.c due to its essen.al oil (Zhang et al.
35 2020; Lazzara et al. 2021). Plant extracts are widely used in the treatment of mild and moderate
36 depression. With the increasing knowledge about the pharmacological ac.vity of plant extracts,
37 hypericin, hyperforin, and similar compounds, interest in St. John's wort has increased.”
This sounds a bit dull. Engage the reader by providing informaon on new insights into
pharmacological advances in the applicaons of Hypericum perforatum, such as its neuroprotecve,
andemenve, or ancancer acvies. Focus parcularly on Hypericin and Hyperforin, as these
compounds are central to this study.
Check those publicaons:
DOI: 10.1016/j.brainres.2017.08.02; 10.1016/j.biopha.2017.05.022; 10.1055/s-2005-873127; 10.1055/s-0033-1351019
In lines 35 to 38, references are missing, and overall, there are many instances where citaons to
support the claims are lacking.
Structure:
Please divide the Materials and Methods secon into headings and formulate it more precisely and
concisely, as this part should be very funconal.
Figures&Tables:
Figure 2 needs to be completely revised. Are there significant differences between the accessions? If
so, they should be represented here. The scale is not labeled and shows different dimensions.
Essenally, differences in properes with smaller values cannot be disnguished. The capon needs
to be more detailed. Comparing the years in a stacked manner does not make sense, as the yearly
differences cannot be visually recognized. Side-by-side comparison would be be2er. Most measured
properes cannot be visually compared because the scale is inappropriate. The quality of the image
should be improved. In general, we recommend not using Excel for creaon but other tools such as R, matplotlib, ggplot, etc.
Figure 1 is a nice illustraon. However, the capons should be more detailed. What can be seen in the different images during seedling producon? Can the dpi of the images be improved?
Although the European Pharmacopoeia specifies a maximum percentage for hyperforin, in Table 5,
the hyperforin content should be given in grams per weight to match hypericin and pseudohypericin
for consistency.
The design of the tables looks good and can be retained as is.
Raw Data are supplied.

Experimental design

2. Experimental design
The present paper is within the scope of the journal. The research question leans more towards
applied scientific/developmental inquiry, describing various genetic variants of Hypericum
perforatum that could be utilized in further programs. The effort of data collection over multiple
years is indeed valuable and should not be dismissed lightly. However, in our opinion, the depth of
the insights gained from the data is not sufficient.
The recorded phenotypic traits seem somewhat randomly chosen. We are compelled to ask why
these particular traits were recorded. What is the intended demonstration?
Significant improvement in the analyses could be achieved through multivariate analysis. Clarifying
the relationships between the traits could provide interesting insights.
To make the results useful for the international research community, serious consideration should be
given to the genetic sequencing of the accessions. While a genome-wide association study is likely
unfeasible due to the number of accessions, a phylogenetic analysis of the main results and the
description of variability would significantly enhance the quality of the paper. If these data have
already been collected elsewhere, they should be cited, referenced, and discussed.
The weather data collected is difficult to interpret due to the single location. Nevertheless, a brief
discussion of possible effects and resilience on the different accessions would be interesng and
would enhance the paper.

Validity of the findings

3. VALIDITY OF THE FINDINGS
The data in the table were not displayed as means—even though this might seem obvious, it should
be indicated. Variance or standard deviation should be included in all tables where means are
provided. Additionally, it should be checked whether the variances are heterogeneous and whether a
mixed model analysis might be preferable to ANOVA in this case. This was not discernible due to the
unlisted variances/standard deviations.
The total number of plants per square meter or simply the total number per block should be
specified. For the figures and tables, it would be interesng to include an "n=..." notaon for each
listed trait or similar data.
The HPLC chromatograms should be briefly presented in the Results secon, including an example
chromatogram of both references and a chromatogram of a sample.

Additional comments

4. General comment
The fundamental idea of the paper to provide a comprehensive description of genetic variants of crop
plant accessions is commendable. These extensive investigations form the basis for deeper analyses
of physiological relationships, environment-genotype interactions, and provide a solid foundation for
breeding programs. However, for a substantial scientific contribution, there is a need for reflection on
the data and deeper analysis (multivariate exploration of the dataset) and contextualization, as well
as integrating the phenotypic data into a genetic context to make the data useful for the scientific
community. If these points are corrected and the overall quality of figures and data display is
significantly improved, we can support acceptance.

Annotated reviews are not available for download in order to protect the identity of reviewers who chose to remain anonymous.

Reviewer 2 ·

Basic reporting

The work “Determination of Natural Populations to be Included in Breeding Program in St. John's Wort Species (Hypericum perforatum L.)” envisioned offering guidelines to pinpoint relevant properties of natural populations of H. perforatum as baseline for a breeding pipeline targeting cultivars with high hypericin and hyperforin that can meet market demands. Overall, this is an accurate and pertinent work, well written, and up to date in the sense that highlights key findings. However, before commending acceptance, I have the following moderate suggestions.

First, I recognize the authors for mentioning in L19 the study aim. However, please also report before the goal statement the research gap, and after the goal statement mention why the proposed characterization is needed and sufficient to bridge the identified gap. Once a broader research gap is presented also in L58, authors can move forward to enlisting not only the study goals, but also research questions, hypothesis and expected results, the latter three still missing from the closing introductory paragraph.

Experimental design

Second, since authors have seed traceability (L81), a key missing genetic parameter worth measuring as baseline for a breeding program is trait heritability. Please follow the recent work at The Forestry Chronicle 2023 99(1):53-66 in order to compute between family variation and contrast it (as a ratio) against overall variation (ca. within family variation + between family variation + error, a model that somehow authors have partly reported under Table 6). This ratio may serve as a proxy for heritability any trait (extend these for all traits currently reported in Figure 2 and Table 4 and 5). Please carry it out and recreate figures 2 and 3 from the suggested paper within this study in order to provide a more robust parameter to guide decision making as part of early breeding phases, such as trait and base-population definition.

Validity of the findings

Third, in terms of results, besides adding new figures as the figures 2 and 3 from The Forestry Chronicle 2023 99(1):53-66, authors must also improve the dendogram already depicted in Figure 2 by adding standard error bars.

Additional comments

Fourth, as closure of the discussion in L174, please include a perspective section before the conclusions section in L178 with recommendation on how to better integrate trait variation from natural populations within the initial phases of H. perforatum breeding. These recommendations would be useful for readers considering expanding the lessons from the H. perforatum. For sure future studies would benefit from these fresh innovative perspectives. As part of this future visions, authors may consider the value of computing general and specific combining abilities for target traits (as done in The Forestry Chronicle 2023 99(1):53-66), as well as the benefit of inter-specific crosses with other Hypericum species to leverage key traits. I would recommend expanding this statement to backcrossing schemes and moder genomic-assisted backcrossing capable to integrated superior seed traits from exotic ancestries in the natural populations into more elite varietal backgrounds (I encourage referring to the recent review Front Genet 2022 13:910386).

---

## Round 0.2 · Minor Revisions

Dear Author,
the MS has certainly improved but a reviewer highlights further very minor suggestions that could be resolved. I kindly invite you to make the requested changes or, alternatively, to justify why such changes are not appropriate.
I'm waiting to receive the new version of this MS.

Best regards

Reviewer 2 ·

Basic reporting

Authors have offered an amended version of the manuscript with a detailed rebuttal. I consider the work has substantially improved.

As last minor suggestions, I do have the following recommendations. When mentioned wild crop relatives in L239, authors may want to mention Front Plant Sci 2023 14:1149469 and Genes 2022 13(1):1.

Additionally, please include a scale bar in the pictures of Figure 1, and report standard errors around the means for tables 6 and 8-10.

Experimental design

See previous review report for well-implemented recommendations.

Validity of the findings

See previous review report for well-implemented recommendations.

Additional comments

NA

---

## Round 0.3 · accepted · Accept

Dear Author

All the last requests of the reviewer have been resolved and I have no further comments therefore, the MS can be accepted for publication.